# Impacts of Rice–Rape Rotation on Major Soil Quality Indicators of Soil in the Karst Region

**DOI:** 10.3390/ijerph191911987

**Published:** 2022-09-22

**Authors:** Hui Fang, Qiuxiao Yan, Zhenming Zhang, Daoping Wang, Xianfei Huang

**Affiliations:** 1Key Laboratory of Chemistry for Natural Products, Guizhou Medical University, Guiyang 550025, China; 2Beautiful Village Construction Center of Quzhou City, Quzhou 324000, China; 3College of Resources and Environmental Engineering, Guizhou University, Guiyang 550025, China; 4Guizhou Provincial Key Laboratory for Environment, Guizhou Normal University, Guiyang 550009, China

**Keywords:** heavy metals in soil, migration and transformation, activity, rice–rape rotation

## Abstract

Arable land resources in karst regions are relatively scarce. The original crop rotation pattern can no longer meet the requirements of productivity development, while different crop rotation patterns have different impacts on the physicochemical properties of the soil. Through field experiments and laboratory analysis, the physicochemical properties and pollution characteristics of the soil during different crop growing stages in rice–rape rotation were investigated systematically. The main results are as follows. During the rice–rape rotation, fine sand in the topsoil experienced the greatest variation. During the rotation, pH variation in the subsoil was greater than that in the topsoil. The soil in paddy fields was poorly ventilated, and the rotation could reduce the redox potential of the soil. In the rotation process, the soil organic matter in the topsoil was higher than that in the subsoil, but the variation of soil organic matter in the topsoil was lower than that in the subsoil. The worst Cd pollution of the topsoil occurred in the seedling stage of rice, while that of the subsoil occurred in the flowering stage of rape; the comprehensive pollution index of Cr and Cd in the subsoil was higher than that in the topsoil. It is of great significance to investigate efficient crop rotation patterns under the conditions of the current productivity for promoting sustainable increases of rape and rice yield, maintaining soil fertility, and improving the soil.

## 1. Introduction

The rice–xerophytic crop rotation system is one of the major crop planting systems in China, and its most remarkable feature is that the alternate changes in hydrothermal conditions of the soil make this system distinct from dry land or wetland ecosystems in terms of the physicochemical properties of the soil, matter circulation, and energy flow, leading to a special ecosystem [1]. Due to the frequent hydroponic operations, the soil structure of chronically flooded paddy fields is damaged, and the redox potential of the soil is reduced, which may cause secondary gleization and influence the growth of rice roots [2]. The implementation of a rice–xerophytic crop rotation would lead to an increase in the number of aggregates and non-capillary pores in the soil, redox potential, and soil permeability, and thus could effectively prevent secondary gleization and acidification of the soil [3,4,5]. Research has shown that, compared to continuous cropping, a rice–xerophytic crop rotation can improve other physicochemical properties of the soil, such as reducing the soil bulk density, increasing the non-capillary porosity, increasing the gas–liquid ratio, and improving the soil pH [6]. In addition, different tillage conditions would lead to different heavy metal activity in the soil, different distributions of plant root systems in the soil, and different absorption levels of heavy metals by crops, and finally influence the accumulation and distribution of heavy metals in the soil [7,8]. Therefore, a study of the physicochemical properties of the soil, nutrient limiting factors, and heavy metal distribution in the rice–xerophytic crop rotation system could play a positive role in improving the crop yield, the fertilization effect, and the utilization rate of fertilizers [9].

There are a great variety of rice–xerophytic crop rotation patterns. Depending on the species of xerophytic crops, the common patterns of rice–xerophytic crop rotation include wheat–rice rotation, rice–rape rotation, green manure–rice rotation, and vegetable–rice rotation [10,11,12]. Guizhou Province, China, is located in the core region of the karst area in southwestern China (KASC) [13,14]. Due to the special geological and geomorphological features of karst areas, these areas are characterized by fewer soil-forming materials and lean and intermittent soil, and arable land resources for agricultural use are limited. In the karst area, the acreage of crops in rice–rape rotation is the largest among the rice–xerophytic crop rotation patterns. Currently, relevant studies on rice–xerophytic crop rotation have mainly focused on simplex aspects such as soil nutrients, physical properties of the soil, and heavy metals in the soil, and most of them were carried out in non-karst areas. However, the wet–dry cycling in a rice–xerophytic crop rotation would cause oxidation and reduction in the soil as well as changes in the soil quality characteristics and the heavy metal distribution. Rice-rape rotation pattern is the most common rotation pattern in karst areas. Systematically studying the influence of human activities on soil physicochemical properties and transformation in soils and accumulation in different organs of metals is of great importance to scientifically manage agricultural activities. In addition, the soil structure of cultivated land in Guizhou Province is complex. Is the accumulation pattern of heavy metals in rice–rape rotation consistent with that in other regions? In our study, the main objectives were to (1) study the changes of soil physical and chemical properties in different crop growth stages of rice–rape rotation; (2) investigate the distribution differences of heavy metals in the soil rice–rape systems at different periods; and (3) identify potential health risks of heavy metals, i.e., with the aim of providing a basis for efficient use of soil fertility and control of heavy metal pollution in soil under the system of rice–rape rotation in a karst region, and promoting food security and the sustainable development of modern agriculture.

## 2. Materials and Methods

### 2.1. The Experiment Material

Shiban Town (27°30′11″ N, 106°44′28″ E), Bozhou District, Zunyi City, Guizhou Province, was selected as the study area (approximately 450 m^2^), which has no factory in here. Youyan No. 57 was selected as the research target and purchased from the Research Institute of Agricultural Sciences of Mianyang through combining Yixiang 1A with the restorative Mianhui 725.

### 2.2. Data Source and Sample Analysis

The local traditional farming method was carried out on the rice–rape rotation (seeding in October and harvesting in May of the following year). In the seedling stage, bolting/tillering stage, filling stage, and harvesting stage of the rape and the rice, the topsoil was collected to determine their concentrations of heavy metals. The experimental methods and processes, including heavy metal analysis, pH value, and organic matter contents of soil samples, were referred to in the previous study [15]. The heavy metals in the soil referred to in the national standard of the Soil Environment Standards (GB15618-2018) was used to evaluate exceeding the standard of heavy metals in soil. Before the rice–rape rotation, we also measured the physical and chemical properties of the soil (Table 1 and Table 2).

Rice was raised in late April 2019, transplanted in early June 2019, and harvested in mid-September 2019. Benzyl ethyl herbicide was used for weeding one week after rice transplanting. Rape was sown in mid October 2019 and harvested at the end of May 2020. Samples of topsoil (0−20 cm) and subsoil (20–40 cm) in the root zone were collected before rapeseed sowing and at the four growth stages (seedling, bolting, flowering, and harvesting) before rice sowing and at the four growth stages (before sowing, seedling, tillering, filling, and harvesting). Five sampling points were set for each sampling unit. Each sampling unit collects three soil samples and mixes them into one sample, and the sampling points were kept away from ridges and roads as far as possible. Each soil sample was marked with sample number, sampling time, location, sampling person, etc. When collecting rice and rape samples, the plant sampling points correspond to the soil sampling points one by one, in each soil sampling unit.

### 2.3. Evaluation of Heavy Metals in Soils

The single pollution index method combined with a comprehensive assessment method and the N. L. Nemerow comprehensive index method were used to assess the heavy metals pollution in soil [16]. The two evaluation methods of heavy metals in soils were referred to in the previous study [17].

## 3. Results

### 3.1. Changes in the Physiochemical Properties of the Soil during Rice–Rape Rotation

#### 3.1.1. Changes in the Mechanical Composition

As shown in Table 3 (the mechanical composition of the topsoil during the rice–rape rotation), the coarse silt accounted for the highest proportion (26.43–35.21%, except for the harvesting stage of rape); coarse and medium sand accounted for the smallest proportion (1.46–2.64%), and there was little difference between the proportions of fine silt, medium silt, and clay particles. Fine silt and medium silt experienced relatively small variations during the rotation (15.12% and 8.44%, respectively), while fine sand experienced the greatest variation during the rotation (variation coefficient: 36.96%) and coarse silt experienced the smallest variation (variation coefficient: 7.94%).

As for the mechanical composition of the subsoil during the rice–rape rotation, all particle sizes (except fine silt, with a variation coefficient of 9.87%) showed a relatively large variation, with find sand experiencing the greatest variation (variation coefficient: 82.91%), followed by clay particles (variation coefficient: 30.44%). Changes in the mechanical composition of the subsoil are closely related to the growth of the crop roots and the activities of the organisms in the soil.

#### 3.1.2. Changes in pH

In the rape growing process, the pH value of the topsoil did not change significantly (Figure 1) with an extremely narrow fluctuation range of 7.80–7.92; the pH value of the subsoil changed significantly during this process and showed a downward trend from the pre-sowing stage to the flowering stage and an upturn during the harvesting stage. Before the seedling stage of rape, the pH value of the subsoil was higher than that of the topsoil, and the value became lower than the latter in all of the following stages and did not increase until the rice season. In the rice growing process, the pH value of the subsoil was higher than that of the topsoil. The pH value of the topsoil and the subsoil showed a consistent trend in the rice growth stages, namely, decreasing first, then increasing, and then decreasing again. In general, the pH value of the subsoil changed more significantly than that of the topsoil during the rice–rape rotation process.

#### 3.1.3. Changes in Redox Potential

In all of the growth stages of rape, the redox potential of the topsoil and the subsoil showed a consistent changing trend (Figure 2), generally showing a wave-like increase, reaching a peak in the bolting stage (179.30 mV for the topsoil and 193.3 mV for the subsoil), decreasing during the flowering stage, and then increasing again in the harvesting stage. The redox potential of the topsoil was slightly lower than that of the subsoil. Eh was positive and within the range of 10.20–193.30 mV, indicating a state of moderate reduction. In the rice growing process, the redox potential of the topsoil and the subsoil showed a consistent changing trend from the seedling stage to the harvesting stage, both increasing first, then decreasing slightly and tending to be stable. The difference between the two was greatly reduced from the pre-planting stage (330.30 mV) to the harvesting stage (2.30 mV), and the changing amplitude of the redox potential of the topsoil was greater than that of the subsoil.

Overall, the changing amplitude of the redox potential of the topsoil was greater than that of the subsoil in the rice–rape rotation. The redox potential of the topsoil reached its highest (179.30 mV) level in the bolting stage of rape and the smallest (−478.30 mV) level in the pre-planting stage of rice. The redox potential of the subsoil reached its highest (193.30 mV) level in the bolting stage of rape and the smallest (−196.50 mV) level in the seedling stage of rape. Both the topsoil and subsoil were in a state of moderate reduction throughout the entire rape growing process. Due to the poor permeability of paddy field soil, the redox potential of the soil was negative, and the soil was in a state of strong reduction throughout the entire rice growing process.

#### 3.1.4. Changes in Soil Organic Matter

In the rice–rape rotation, the soil organic matter of the topsoil was higher than that of the subsoil (Figure 3), while the changing amplitude of the soil organic matter of the topsoil was smaller than that of the subsoil. The soil organic matter of the subsoil in the rice growing period was significantly lower than that in the rape growing period. The soil organic matter of the subsoil in the pre-planting stage of rice was 23.37 g kg^−1^ lower than that in the harvesting stage of rape. This may be because the subsoil had long been in an anaerobic environment and might be subjected to a high mineralization rate; meanwhile, the soluble organic carbon and biomass of the soil microorganisms in the paddy field soil would be significantly reduced in flooded conditions. The soil organic matter of the subsoil was reduced again in the tillering stage of the rice, and this decrease may be related to the further downward growth of the rice and its absorption of nutrients from the subsoil in this stage. The soil organic matter of the subsoil increased continuously in the filling stage and harvesting stage. In these stages, most nutrients needed for the growth and development of rice grains are obtained through leaf photosynthesis and from the topsoil, and in-plant transfer and root ageing could also increase the soil organic matter of the subsoil.

### 3.2. Changes in the Heavy Metal Content of the Soil during Rice–Rape Rotation

There are many factors contributing to changes in the heavy metal content of farmland soil. For example, crop growth can enrich heavy metals in plants, while the metals can run off with rainwater. Meanwhile, atmospheric deposition can increase the heavy metal content of the soil. In addition, there are many other factors that can change the heavy metal content of the soil during crop growth.

#### 3.2.1. Changes in the Cr and Cd Content of the Soil

As shown in Figure 4a, the Cr content of the topsoil and subsoil was fluctuating before the tillering stage of rice, but the fluctuation range was relatively narrow. Chromium content in topsoil samples fluctuated within the range from 71.18 to 84.36 mg kg^−1^ (variation coefficient = 4.65%), and Cr content of subsoil samples fluctuated within the range from 77.06 to 89.07 mg kg^−1^ (variation coefficient = 4.38%). Chromium content in both topsoil and subsoil decreased continuously during the period from filling to harvesting stages of rice. In comparison to the tillering stage, Cr content decreased by 25.13 mg kg^−1^ in topsoil and by 35.11 mg kg^−1^ in subsoil in the filling stage. In the harvesting stage, the Cr content of the topsoil and subsoil showed a synchronous decreasing trend. Overall, the Cr content of the subsoil was higher than that of the topsoil (except for in the harvesting stage of rape); the difference in the Cr content between the topsoil and subsoil diminished in the bolting stage of rape, but tended to increase in the subsequent fluctuations, and then diminished again in the filling and harvesting stages of the rice.

As seen in Figure 4b, the Cd content of the topsoil increased first and then decreased during the rape growing process and reached its highest level during the bolting stage (1.73 mg kg^−1^); during the rice growing process, the content showed the same changing trend and reached its highest level in the seedling stage (1.81 mg kg^−1^).

The Cd content of the subsoil decreased first, then increased and decreased again during the rape growing process, while the content increased first and then decreased during the rice growing process. The variation of Cd content in the rice growing process (variation coefficient: 42.38%) was higher than that in the rape growing process (variation coefficient: 22.35%). On the whole, although the Cd content of the topsoil and subsoil in the rice–rape rotation fluctuated, it showed a decreasing trend from the harvesting stage of rape to the pre-planting stage of rice and from the filling stage of rice to the harvesting stage of rice.

#### 3.2.2. Changes in the Pb and Cu Content of the Soil

During the rice–rape rotation, the Pb content of the topsoil and subsoil was generally exhibiting a wave-like rise; the Pb content of the topsoil was higher than that of the subsoil in all of the stages except for the seedling stage and filling stage of rice (see Figure 5a). As seen in Figure 5b, the Cu content of the topsoil increased first and then decreased during the rape growing process, while the changing trend of the Cu content of the subsoil showed the reverse, where the content of both the topsoil and subsoil experienced small variations (with a variation coefficient of 5.67% for the topsoil and a variation coefficient of 4.72% for the subsoil). In the rice growing process, the Cu content of the soil increased first and then decreased, and the variations were significantly higher than those during the rape growing process (with a variation coefficient of 35.33% for the topsoil and a variation coefficient of 30.74% for the subsoil). Overall, the Cu content of the topsoil and subsoil showed a generally consistent changing trend during the rice–rape rotation.

### 3.3. Evaluation of the Soil Pollution by Heavy Metals in Rice–Rape Rotation

According to the Environmental Quality Standard for Soils issued in 1995, the N.L. Nemerow comprehensive index method was used. The single pollution index (Pi) reflecting the heavy metal content of the soil in each growth stage of the rape and rice, as well as the comprehensive pollution index (PZ) reflecting the heavy metal content during the growing process of the rice and rape, were calculated, and the heavy metals in the soil were analyzed and evaluated.

As seen in Table 4, the Cd in both the topsoil and subsoil was at the level of moderate pollution (PZ = 2.70 for the topsoil, PZ = 2.79 for the subsoil), and the subsoil was more polluted by Cd than the topsoil. The worst Cd pollution of the topsoil occurred in the seedling stage of rice, while that of the subsoil occurred in the flowering stage of rape. The other four elements were at safe levels in both the topsoil and subsoil. The single index of heavy metal pollution showed inconsistent changing trends throughout the entire growing periods of rice and rape. However, except for Cr and Cd, the comprehensive pollution index of the other three heavy metals in the subsoil was smaller than that in the topsoil.

## 4. Discussion

The soil mechanical composition is not only an important diagnostic index of soil classification but also a critical factor affecting soil moisture, fertilizer, gas, heat and matter migration and transformation, as well as the soil degradation process [12]. In addition to reflecting the sizes and the number of soil mineral particles, it also has an influence on the performance of the soil, such as farming, seedling, and sowing behaviors, etc. Regardless of rice and rape cultivation, a distribution pattern embodied by a loose upper layer followed by the middle layer and finally the lower layer is presented. For soil immersed in water for a long time, the soil particles are scattered, and the soil texture is poor. However, paddy-upland rotation can correct the adverse effect of the water layer on the soil structure. If the soil is in a flooding condition continuously, the soil oxidation reduction potential declines and the soil crumb structure is damaged, which leads to soil hardening. Consequently, poor soil aeration further results in the deterioration of soil characteristics. Thus, an unfavorable influence affects crop root growth and crop growth [18]. Subsequent to turning a paddy field into dry farming, the soil begins to become loose, and both the oxidization and reduction potential and the number of soil particles increases. In this case, secondary gleization can be removed to provide a good rhizospheric environment for crop growth. In the process of the rice–rape rotation, surface soil oxidization and reduction potential change to a greater extent than in the basement soil. Moreover, both the surface and the basement soil are in a moderate reduction state throughout the whole growth period of rape. Throughout the whole rice growth period, aeration condition of paddy soil is poor and redox potential is negative, and soil is strongly reduced in this time.

The organic substance content in the soil is an essential index, revealing the fertility conditions of a cropland [19,20]. When subjected to traditional farming conditions, frequent turning improves the soil aeration and the contact area between the microorganisms and organic matter is increased. In this way, the organic matter mineralization rate is significantly higher than that during rice–rape rotation. Therefore, the organic matter content in the surface soil during rice–rape rotation is above that in the basement soil, while the variation amplitude of the former is below the latter. In addition, the reduction amplitude of the organic matter in the soil during the growth period of rice is greater than that in the growth period of rape. Then, it is reduced further in the rice filling stage and then returns to a normal level. Under the condition of long-term flooding, the entire tillage layer is in a moderately or strongly reduced state. With the increase in cultivation years, the organic matter content in the soil in such a rice–rape rotation system increases in all cases. As the soil pH values are enhanced, soil acidification is alleviated.

Heavy metal enters cropland by means of atmospheric precipitation, sewage irrigation, and fertilizer and pesticide application, etc., and accumulates in the soil [21,22]. In traditional farming conditions, the heavy metals may be uniformly distributed in the entire tillage layer through ploughing [23]. Furthermore, the reason why the heavy metal absorbing capacity of the soil is different is that heavy metal activities and the plant root system distribution are different in the soil under different farming conditions, which eventually affects the heavy metal accumulation and distribution in the soil [24]. In terms of the heavy metals Cr and Cd, the variation tendencies of their contents in the soil are rather similar to each other during rice–rape rotation. In the rape growth period, but before the rice tillering stage, Cr and Cd contents in the surface soil and basement soil fluctuate within a small range. However, their contents decrease dramatically in the surface soil and the basement soil after the rice tillering stage, and the differences in their contents become less significant in such soil. In the course of rice–rape rotation, heavy metal Pb content in the surface soil and the basement soil is inclined to fluctuate and increase on the whole. Except for the rice seedling and tillering stages, the Pb content in the surface soil is higher than that in the basement soil. For the heavy metal Cu content in the soil, it was rather steady during the growth period of rape, while it increased followed by a decrease during the rice growth stage. The extent of such variations was significantly greater than that in the growth period of rape.

## 5. Conclusions

During the rice–rape rotation, coarse silt accounted for the highest proportion in the topsoil, while coarse and medium sand accounted for the smallest proportion; fine sand in the topsoil experienced the greatest variation, while coarse silt experienced the smallest variation. In the subsoil, all particle sizes (except fine silt that experienced a small variation) showed a relatively large variation. In the rape growing period, the pH variation in the topsoil was insignificant and within the range of 7.80–7.92; the pH value of the topsoil and the subsoil showed a consistent trend during the rice growing stages, namely, decreasing first, then increasing and then decreasing again. During the rotation process, the soil organic matter in the topsoil was higher than that in the subsoil. However, variation of soil organic matter in the two soil layers is opposite. According to the single pollution index and comprehensive pollution index, Cr, Pb, and Cu belonged to the cleanliness level, while Cd belonged to the moderate pollution level. The worst Cd pollution of the topsoil occurred in the seedling stage of rice and that of the subsoil occurred in the flowering stage of rape. The comprehensive pollution index of Cr and Cd in the subsoil was higher than that in the topsoil, while the comprehensive pollution index of Pb and Cu in the subsoil was smaller than that in the topsoil. Cultivated soil is a complex system, and the effects of soil physical and chemical properties on heavy metal activities generally do not play their respective roles. Therefore, the dynamic changes of soil quality should be comprehensively considered in long-term human activities and the cultivation process.

## Figures and Tables

**Figure 1 ijerph-19-11987-f001:**
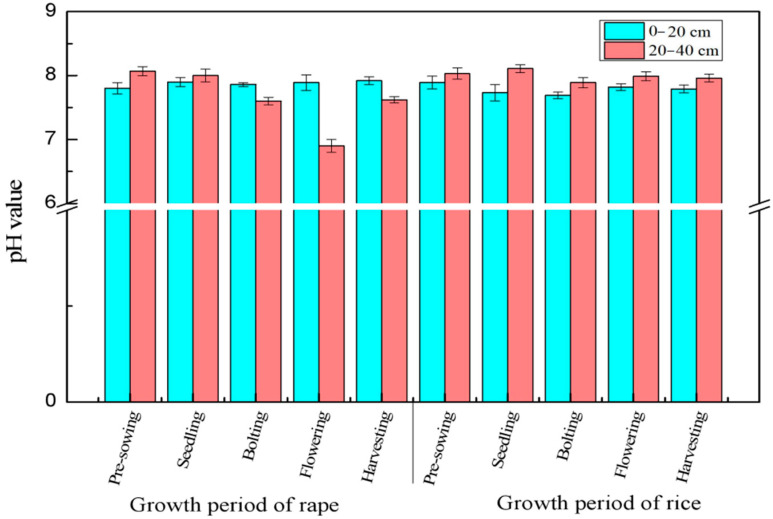
Change of spill pH value during rice–rape rotation.

**Figure 2 ijerph-19-11987-f002:**
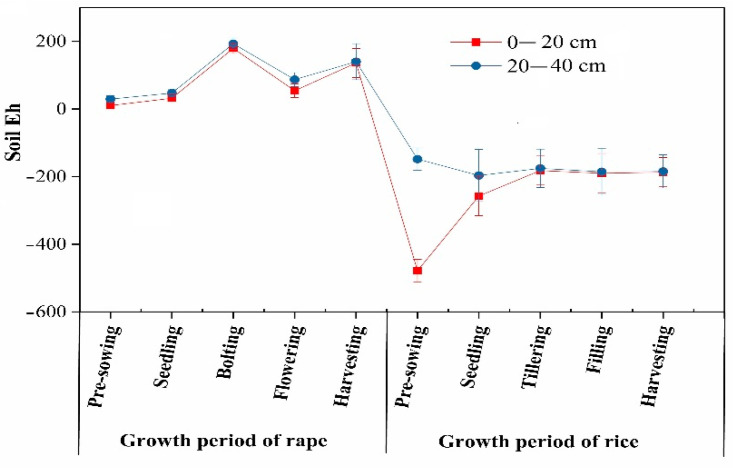
Variation of soil Eh during rice–rape rotation.

**Figure 3 ijerph-19-11987-f003:**
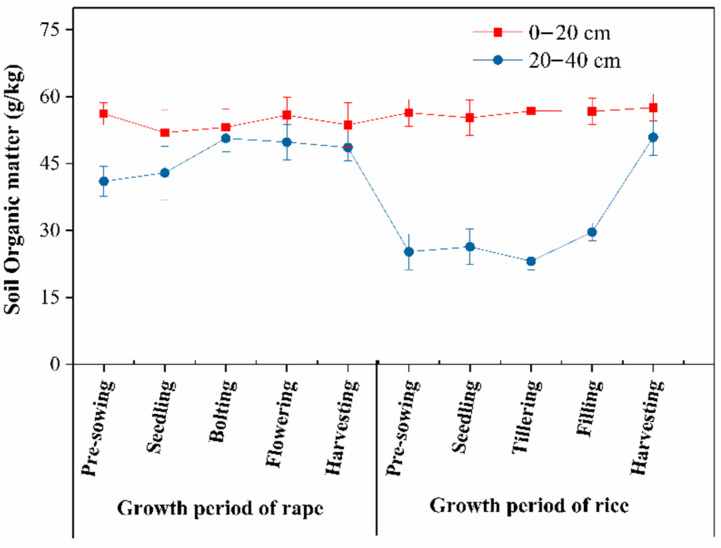
Changes of soil organic matter during rice–rape rotation.

**Figure 4 ijerph-19-11987-f004:**
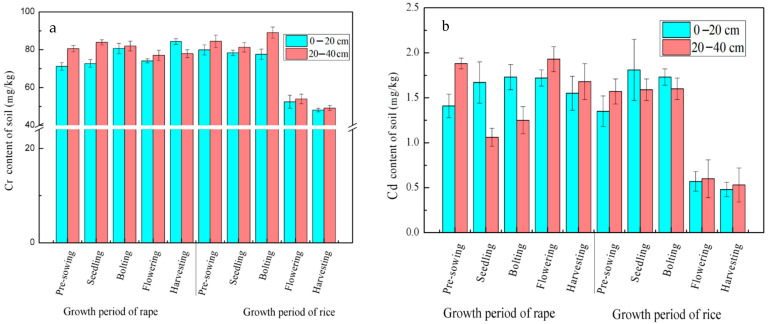
Change of heavy metal Cr (**a**) and Cd (**b**) in soil during rice–rape rotation.

**Figure 5 ijerph-19-11987-f005:**
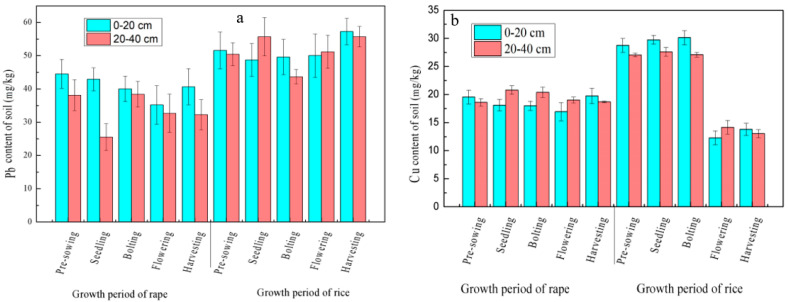
Change of heavy metal Pb (**a**) and Cu (**b**) in soil during rice–rape rotation.

**Table 1 ijerph-19-11987-t001:** Survey of soil chemical indexes in the study area.

Soil Depth	Index	Soil Organic Matter	Eh	pH	Pb	Cu	Zn	Cr	Cd
	Mean	56.19	10.2	7.8	44.54	19.54	81.72	56.19	1.41
0–20 cm	Standard deviation	2.38	12.62	0.09	4.37	1.22	1.23	2.38	0.13
	Coefficient of variation (%)	4.24	123.73	1.15	9.81	6.24	1.51	4.24	9.22
	Mean	41.03	29.2	8.07	38.14	18.59	69.49	80.61	1.88
20–40 cm	Standard deviation	2.64	6.72	0.1	4.63	0.69	0.57	1.58	0.06
	Coefficient of variation (%)	6.43	23.01	1.24	12.14	3.71	0.82	1.96	3.19

**Table 2 ijerph-19-11987-t002:** Composition of soil particles in the study area (unit: %).

Particle Size	0–20 cm	20–40 cm
Clay particles	18.97	23.85
Fine silt	12.43	16.76
Medium silt	15.81	16.44
Coarse silt	35.21	34.06
Fine sand	14.96	6.64
Coarse and medium sand	2.64	2.26

**Table 3 ijerph-19-11987-t003:** Changes of soil machinery during rice–rape rotation (unit:%).

Soil Depth	Particle Size	Growth Period of Rape					Growth Period of Rice					Average Value	Coefficient of Variation
		Pre-Sowing	Seedling	Bolting	Flowering	Harvesting	Pre-Sowing	Seedling	Tillering	Filling	Harvesting		
0–20	Clay particles	18.97	17.25	24.46	11.16	10.26	19.76	17.68	16.69	18.51	16.43	17.12	22.61
	Fine silt	12.43	14.98	18.11	16.23	16.37	13.42	16.12	21.37	17.37	19.14	16.55	15.12
	Medium silt	15.81	17.87	18.62	17.55	15.23	20.49	19.66	17.52	17.37	18.51	17.86	8.44
	Coarse silt	35.21	32.53	31.82	33.67	26.43	30.37	30.99	31.15	29.74	27.87	30.98	7.94
	Fine sand	14.96	15.25	5.47	19.28	29.59	14.44	13.67	11.81	15.13	16.49	15.61	36.96
	Coarse and medium sand	2.64	2.13	1.51	2.11	2.11	1.53	1.88	1.46	1.88	1.56	1.88	19.03
20–40	Clay particles	23.86	17.28	30.55	14.32	10.52	30.06	30.89	30.42	27.87	20.28	23.60	30.44
	Fine silt	16.76	15.64	19.11	16.36	21.65	20.49	17.37	17.21	18.72	18.82	18.21	9.87
	Medium silt	16.44	17.68	19.21	15.65	10.52	27.87	19.45	18.15	16.22	15.91	17.71	23.40
	Coarse silt	34.06	31.59	28.72	33.14	34.83	16.43	28.70	30.63	29.12	30.37	29.76	16.47
	Fine sand	6.64	14.05	0.79	18.22	20.26	2.95	1.21	1.68	5.39	12.83	8.40	82.91
	Coarse and medium sand	2.26	3.76	1.61	2.31	2.23	2.20	2.38	1.91	2.67	1.79	2.31	24.37

**Table 4 ijerph-19-11987-t004:** Evaluation of soil heavy metal pollution during rice–rape rotation.

Soil Depth	Element	Growth Period of Rape (P*_i_*)					Growth Period of Rice (P*_i_*)					P*_Z_*	Class of Pollution
		Pre-Sowing	Seedling	Bolting	Flowering	Harvesting	Pre-sowing	Seedling	Tillering	Filling	Harvesting		
0–20	Cr	0.28	0.29	0.31	0.30	0.34	0.32	0.31	0.31	0.21	0.19	0.31	Security
	Cd	2.35	2.78	2.88	2.86	2.59	2.26	3.01	2.88	0.95	0.79	2.70	Moderate pollution
	Pb	0.13	0.12	0.11	0.10	0.12	0.15	0.14	0.14	0.14	0.16	0.15	Security
	Cu	0.20	0.18	0.18	0.17	0.20	0.29	0.30	0.30	0.12	0.14	0.26	Security
	Zn	0.27	0.25	0.27	0.24	0.27	0.25	0.25	0.25	0.25	0.25	0.26	Security
20–40	Cr	0.32	0.34	0.33	0.31	0.31	0.34	0.33	0.36	0.22	0.20	0.33	Security
	Cd	3.14	1.76	2.09	3.22	2.81	2.62	2.64	2.66	1.01	0.88	2.79	Moderate pollution
	Pb	0.11	0.07	0.11	0.09	0.09	0.14	0.16	0.12	0.15	0.16	0.14	Security
	Cu	0.19	0.21	0.20	0.19	0.19	0.27	0.28	0.27	0.14	0.13	0.24	Security
	Zn	0.23	0.26	0.25	0.24	0.23	0.23	0.23	0.25	0.24	0.26	0.25	Security

## Data Availability

The data presented in this study are available on request from the corresponding author.

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
