# Peer review of "Impacts of Rice–Rape Rotation on Major Soil Quality Indicators of Soil in the Karst Region"

_ijerph, 2022, doi:10.3390/ijerph191911987_

Round 1
Reviewer 1 Report
The reviewer greatly thanks the Editor for having allowed reviewing this manuscript. The review paper ijerph-1891650entitled " Impacts of rice-rape rotation on major soil quality indicators of soil in the Karst region" is attractive and has information relevant to the audience. The review paper is generally well written and structured. Moreover, the authors provide information about the status of rice-rape rotation effects on major soil quality indicators of soil in the Karst region and the lack of information on this topic. However, some revisions must be conducted before its acceptance. Here are some general comments:
General comments
1. The authors must compare their results with existing works without rice-rape rotation.
2. Some grammatical errors and punctuation must be fixed and polished.
3. Lines 57-58: please add recent references.
4. The authors should sufficiently explain the paper's novelty in the introduction section before stating the work’s objectives.
5. Lines 175-178, 192-193, 301-303, etc.: some texts and data overlap with other published articles (e.g. www.pjoes.com). By this, rewrite or explain this assertion.
6. To «conclusion», add perspectives. Try to provide some gaps for further research; this will enhance the novelty of the paper.
Author Response
Dear editor and reviewers,
Firstly, thanks a lot to the contribution by the editors and reviewers. All the comments are precious and indispensable for the perfect of this manuscript and will help to improve the importance of our research.
Based on the comments of reviewers and your suggestions, I have carried out the corresponding revision for my paper. The replies to referees are as follow one by one. In addition, all the amendments in this revised paper are provided with a Blue font so that you could check clearly.
We have revised the manuscript based on the comments from editors and reviewers as followed:
- The authors must compare their results with existing works without rice-rape rotation.
Reply: Thanks a lot to the contribution by the reviewer. We have added soil physical and chemical indicators before this study(See Table 1 and table 2)Since the area of this study is a long-term study on the rotation of rice and rapeseed, we have also slightly added the corresponding content in the discussion for description and comparison.
- Some grammatical errors and punctuation must be fixed and polished.
Reply: We have revised the grammar and punctuation of the full text, and have asked senior experts of our profession to help check it. Thanks!
- Lines 57-58: please add recent references.
Reply: Thanks a lot to the contribution by the reviewer. We have added references to the manuscript. (Lines 57-58)
- The authors should sufficiently explain the paper's novelty in the introduction section before stating the work’s objectives.
Reply: Thanks a lot to the contribution by the reviewer. In the introduction, we have made major revisions and supplemented the research objectives and innovative statements of this manuscript.
- Lines 175-178, 192-193, 301-303, etc.: some texts and data overlap with other published articles (e.g. www.pjoes.com). By this, rewrite or explain this assertion.
Reply: Thank you very much for your suggestions. Lines 175-178, 192-193, 301-303, et have all been revised. Other corresponding contents have also been written from the manuscript.
6、To «conclusion», add perspectives. Try to provide some gaps for further research; this will enhance the novelty of the paper.
Reply: Thanks a lot to the contribution by the reviewer. We have revised the conclusion and added new viewpoints of the manuscript.
If you have any question about this paper, please don`t hesitate to let me know.
Thank you and all the referees very much for the kind advice.
Sincerely yours,
Qiuxiao Yan
Key Laboratory of Chemistry for Natural Products, Guizhou Medical University
Tel: +86-15285647805
E-mail: yanqxecho@sina.com

Reviewer 2 Report
It is not clear that how long ( year/s) this crop rotation has been adopted.
References cited in 58, 87 and 94 are miss match with detail Reference ( year)
93- author name is not correct
99- 26.43% - 35.21 instead of 6.43% - 35.21%
112- Table No. 1 instead of Table No. 3
234- Table No. 2 instead of Table No. 4
256 & 257 may be page no missing
Author Response
Dear editor and reviewers,
Firstly, thanks a lot to the contribution by the editors and reviewers. All the comments are precious and indispensable for the perfect of this manuscript and will help to improve the importance of our research.
Based on the comments of reviewers and your suggestions, I have carried out the corresponding revision for my paper. The replies to referees are as follow one by one. In addition, all the amendments in this revised paper are provided with a Blue font so that you could check clearly.
We have revised the manuscript based on the comments from editors and reviewers as followed:
1、It is not clear that how long ( year/s) this crop rotation has been adopted.
Reply: Thanks a lot to the contribution by the reviewer. We have modified and supplemented the corresponding contents in the materials and methods section. This research area has always been in rice-rape rotation, and we also supplemented the start time of the experiment.
2、 References cited in 58, 87 and 94 are miss match with detail Reference ( year)
Reply: We have added the missing references. Thanks!
3、93- author name is not correct
Reply: We have revised the corresponding references. Thanks!
4、99- 26.43% - 35.21 instead of 6.43% - 35.21%
Reply: Thanks a lot to the contribution by the reviewer. We have revised the corresponding content. Thanks!
5、112- Table No. 1 instead of Table No. 3
Reply: We have revised the corresponding content. Thanks!
6、234- Table No. 2 instead of Table No. 4
Reply: We have revised the corresponding content. Thanks!
7、256 & 257 may be page no missing
Reply: We have revised the corresponding content. Thanks!
If you have any question about this paper, please don`t hesitate to let me know.
Thank you and all the referees very much for the kind advice.
Sincerely yours,
Qiuxiao Yan
Key Laboratory of Chemistry for Natural Products, Guizhou Medical University
Tel: +86-15285647805
E-mail: yanqxecho@sina.com

Reviewer 3 Report
Dear Authors,
The manuscript “Impacts of rice-rape rotation on major soil quality indicators of soil in the Karst region” introduces to readers some information about changes in physiochemical properties of soil in the rice-rape rotation system. Unfortuanately, I feel sorry to say that the paper is not good enough to publish. I suggest authors should rewrite the paper again with more detailed in some points.
Method
For experiment design: The paper did not mention how to disign field exxperiment, for example: Time for each crop? Treatments/how many replicates? Although authors mentioned some methods were in Zhang et al. 2019, but that paper for rice-wheat??
Because there was no information about replicates, I could not trust on data and some data had no error bar. The paper had no statistic analysis? Then, how can you conclude that was significant or not (for most figures and table).
Figure 3: Soil organic matter: Beside there was lack of error bars, I put a question on values of SOM of growth period of rice in the soil layer 20-40: there was 20mg/kg different in some periods: equal to 40 Mg ha-1: how it could be? Or errors?
And there is meaningless when comparing the physicochemical properties between the top soil and sub-soil, which paper has focused on.
Author Response
Dear editor and reviewers,
Firstly, thanks a lot to the contribution by the editors and reviewers. All the comments are precious and indispensable for the perfect of this manuscript and will help to improve the importance of our research.
Based on the comments of reviewers and your suggestions, I have carried out the corresponding revision for my paper. The replies to referees are as follow one by one. In addition, all the amendments in this revised paper are provided with a Blue font so that you could check clearly.
We have revised the manuscript based on the comments from editors and reviewers as followed:
1、For experiment design: The paper did not mention how to disign field exxperiment, for example: Time for each crop? Treatments/how many replicates? Although authors mentioned some methods were in Zhang et al. 2019, but that paper for rice-wheat??
Reply: Thanks a lot to the contribution by the reviewer. We have revised the experimental design in the manuscript. The details are as follows:
Rice was raised in late April 2019, transplanted in early June 2019, and harvested in mid September 2019. Benzyl ethyl herbicide was used for weeding one week after rice transplanting. Rape was sown in mid October 2019 and harvested at the end of May 2020. Samples of topsoil (0-20cm) and subsoil (20-40cm) in the root zone were collected before rapeseed sowing and at the four growth stages (seedling, bolting, flowering and harvesting), before rice sowing and at the four growth stages (before sowing, seedling, tillering, filling and harvesting). Five sampling points were set for each sampling unit, Each sampling unit collects three soil samples and mixes them into one sample and the sampling points were kept away from ridges and roads as far as possible. Each soil sample was marked with sample number, sampling time, location Sampling person, etc. When collecting rice and rape samples, the plant sampling points correspond to the soil sampling points one by one, in each soil sampling unit.
- Because there was no information about replicates, I could not trust on data and some data had no error bar. The paper had no statistic analysis? Then, how can you conclude that was significant or not (for most figures and table).
Reply: Thanks a lot to the contribution by the reviewer. Regarding the research data of this article, we uploaded the data as an attachment. At the same time, we have modified the tables and drawings. The tables add errors and coefficients of variation, and Figure 2 and figure 3 add error analysis.
3、Figure 3: Soil organic matter: Beside there was lack of error bars, I put a question on values of SOM of growth period of rice in the soil layer 20-40: there was 20mg/kg different in some periods: equal to 40 Mg ha-1: how it could be? Or errors?
Reply: Thanks a lot to the contribution by the reviewer.Figure 3 we have modified and added error analysis. In the manuscript we analyze this problem, the details are as follows:
“This may be because the subsoil had long been in an anaerobic environment and might be subjected to a high mineralization rate; meanwhile, the soluble organic carbon and biomass of the soil microorganisms in the paddy field soil would be significantly reduced in flooded conditions. The soil organic matter of the subsoil was reduced again in the tillering stage of the rice, and this decrease may be related to the further downward growth of the rice and its absorption of nutrients from the subsoil in this stage. The soil organic matter of the subsoil increased continuously in the filling stage and harvesting stage. In these stages, most nutrients needed for the growth and development of rice grains are obtained through leaf photosynthesis and from the topsoil, and in-plant transfer and root ageing could also increase the soil organic matter of the subsoil. ”
4、And there is meaningless when comparing the physicochemical properties between the top soil and sub-soil, which paper has focused on.
Reply: Thanks a lot to the contribution by the reviewer. Due to human activities and different farming methods, and the root system of rice is different from that of rape As a result, the physical and chemical properties of the surface and bottom soils are different. Based on these considerations, we study the physical and chemical properties of the surface and bottom soils.
If you have any question about this paper, please don’t hesitate to let me know.
Thank you and all the referees very much for the kind advice.
Sincerely yours,
Qiuxiao Yan
Key Laboratory of Chemistry for Natural Products, Guizhou Medical University
Tel: +86-15285647805
E-mail: yanqxecho@sina.com
